# Discovery of an Insect Neuroactive Helix Ring Peptide from Ant Venom

**DOI:** 10.3390/toxins15100600

**Published:** 2023-10-05

**Authors:** Valentine Barassé, Laurence Jouvensal, Guillaume Boy, Arnaud Billet, Steven Ascoët, Benjamin Lefranc, Jérôme Leprince, Alain Dejean, Virginie Lacotte, Isabelle Rahioui, Catherine Sivignon, Karen Gaget, Mélanie Ribeiro Lopes, Federica Calevro, Pedro Da Silva, Karine Loth, Françoise Paquet, Michel Treilhou, Elsa Bonnafé, Axel Touchard

**Affiliations:** 1EA-7417, Institut National Universitaire Champollion, Place de Verdun, 81012 Albi, France; 2Centre de Biophysique Moléculaire, Centre National de la Recherche Scientifique (CNRS), Unité Propre de Recherche (UPR) 4301, 45071 Orléans, France; 3Unité de Formation et de Recherche (UFR) Sciences et Techniques, Université d’Orléans, 45071 Orléans, France; 4Inserm, Univ Rouen Normandie, NorDiC Unité Mixte de Recherche (UMR) 1239, 76000 Rouen, France; 5Laboratoire Écologie Fonctionnelle et Environnement, Université de Toulouse, CNRS, Toulouse INP, Université Toulouse 3-Paul Sabatier (UPS), 31062 Toulouse, France; 6Unité Mixte de Recherche (UMR) Écologie des Forêts de Guyane (EcoFoG), AgroParisTech, Centre de Cooperation Internationale en Recherche Agronomique pour le Développement (CIRAD), Centre National de la Recherche Scientifique (CNRS), Institut National de Recherche pour l’Agriculture, l’Alimentation et l’Environnement (INRAE), Université des Antilles, Université de Guyane, 97379 Kourou, France; 7Institut National de Recherche pour l’Agriculture, l’Alimentation et l’Environnement (INRAE), Institut National des Sciences Appliquées (INSA) de Lyon, Biologie Fonctionnelle, Insectes et Interactions (BF2i), Unité Mixte de Recherche (UMR) 203, Université de Lyon, 69621 Villeurbanne, France

**Keywords:** ant venoms, neurotoxins, *Tetramorium bicarinatum*, potassium channel, helix peptide

## Abstract

Ants are among the most abundant terrestrial invertebrate predators on Earth. To overwhelm their prey, they employ several remarkable behavioral, physiological, and biochemical innovations, including an effective paralytic venom. Ant venoms are thus cocktails of toxins finely tuned to disrupt the physiological systems of insect prey. They have received little attention yet hold great promise for the discovery of novel insecticidal molecules. To identify insect-neurotoxins from ant venoms, we screened the paralytic activity on blowflies of nine synthetic peptides previously characterized in the venom of *Tetramorium bicarinatum*. We selected peptide U_11_, a 34-amino acid peptide, for further insecticidal, structural, and pharmacological experiments. Insecticidal assays revealed that U_11_ is one of the most paralytic peptides ever reported from ant venoms against blowflies and is also capable of paralyzing honeybees. An NMR spectroscopy of U_11_ uncovered a unique scaffold, featuring a compact triangular ring helix structure stabilized by a single disulfide bond. Pharmacological assays using *Drosophila* S2 cells demonstrated that U_11_ is not cytotoxic, but suggest that it may modulate potassium conductance, which structural data seem to corroborate and will be confirmed in a future extended pharmacological investigation. The results described in this paper demonstrate that ant venom is a promising reservoir for the discovery of neuroactive insecticidal peptides.

## 1. Introduction

One of the major challenges in agriculture today is the development of novel biological agents to control arthropod pests. Global changes (e.g., monocultures, climate change, introduction of invasive exotic species) favor the proliferation of insect pests, while the extensive use of synthetic and persistent pesticides to fight these harmful insects has triggered resistance. Additionally, the insecticidal molecules currently in use are widely recognized as one of the main factors responsible for the worldwide decline in biodiversity, and as a notable risk for human health. The negative impacts of these pesticides have forced the European Union to severely restrict their use (e.g., 183 conventional insecticides were deregistered between 1993 and 2008 [1]), hence limiting the diversity of pesticides currently available. The decline of registered synthetic insecticides consequently leads to unmet agronomic needs for crops and seed storage. Therefore, it becomes paramount to develop an arsenal of new eco-friendly pesticides as alternatives. Peptides are considered as having negligible impacts on the environment since they have short half lives and their degradation products are amino acids, which are neither toxic nor persistent. Several predatory arthropods secrete a highly toxic venom mostly composed of peptides to subdue prey [2]. The insecticidal peptides contained in the venoms of such entomophagous predators therefore constitute a pertinent and promising alternative to the synthetic chemicals used by the agrochemical industry. These venom toxins, whose selectivity on the pharmacological receptors of their prey has been fine-tuned through natural selection processes, offer the opportunity to derive bioinsecticides that are not toxic for vertebrates and are specific to certain groups of insects. Moreover, the large diversity of pharmacological sites targeted by venom peptides would allow the development of insecticides with innovative modes of action, limiting the phenomenon of resistance. The potential of venom peptides as bioinsecticides has been exemplified by the commercialization in 2018 of Spear^®^ products (Vestaron, Kalamazoo, MI, USA). The active ingredient in Spear^®^ is the peptide GS-ω/κ-hexatoxin-Hv1a (Hv1a), isolated from the venom of the funnel-web spider *Hadronyche versuta* [3]. The peptide Hv1a is active through ingestion and contact on several insects (e.g., Lepidoptera, spider mites, flies, aphids) and was demonstrated as safe for beneficial pollinators [4]. However, although safe for honeybees, Spear^®^ is toxic to a broad array of arthropods and therefore retains a rather negative impact on biodiversity. Characterizing other insecticidal peptides highly specific for targeted pests is the next major challenge to further reduce the environmental impact of venom-derived bioinsecticides. To increase the possibility of discovering new selective peptides with innovative modes of action for controlling targeted pests, it is necessary to broaden the spectrum of the venomous animals that are explored.

Ants, which dominate most terrestrial environments, are one of the most abundant groups of venomous organisms on Earth, with 14,141 extant species described thus far [5], and are among the leading predators of invertebrates in most ecosystems [6]. To subdue their prey, they have evolved an arsenal of adaptations and weapons, including potent peptidic venoms with paralytic and lethal effects on many types of arthropods [7,8]. Venom peptides represent a vast source of structurally diverse and biologically active compounds with a potentially high potency and narrow selectivity for a range of arthropod species [9]. The discovery of a new structural family of neuroactive peptides in the venom of the Amazonian ant *Anochetus emarginatus* demonstrated the promising potential of this field of research [10]. Nevertheless, apart from this work, very few studies [11,12,13,14,15] have been conducted to characterize the insecticidal activity of the peptide toxins isolated from ant venoms. The main reason for the lack of research on ants is that most species are less than 1 cm long and provide a little amount of venom, which has made peptide sequence identification daunting. Proteo-transcriptomic analyses have recently revolutionized venom peptide sequencing from small venomous organisms [16], including several ant species [17,18,19]. These investigations have begun to reveal the molecular diversity of ant venoms, with many newly discovered peptides, while the major challenge remains the determination of their biological activities.

Since the discovery of the bicarinalin toxin (M-Tb1a), our research group has been characterizing the venom of the ant *Tetramorium bicarinatum* [20]. The complete peptide repertoire of the venom has previously been elucidated through a proteo-transcriptomics approach, which led to the identification of 37 peptides divided into 17 toxin groups (U_1_-U_17_) based on their sequence similarity. Among these peptides, we demonstrated that M-Tb1a and U_9_ (recently renamed M-Tb2a) are cytotoxic by targeting cell membranes [21], and that U_1_ (formerly named P17) is an agonist of the G protein-coupled receptor MRGPRX2, inducing a pro-inflammatory response in mammals [22]. In addition, based on sequence similarity with Ta3a, U_3_ peptide is putatively a voltage-gated sodium channel modulator capable of eliciting pain in vertebrates [23]. We are therefore pursuing the characterization of biological activities and identification of pharmacological targets of other groups of toxins. *Tetramorium* are predatory ant species that employ their venom to prey upon small arthropods [24]. Although insecticidal activities have recently been noted for M-Tb1a and U_9_ [21], there has been no extended investigation of the insecticidal properties of venom peptides from *T. bicarinatum*. In this study, we explored the paralytic and insecticidal activities of a panel of synthesized *T. bicarinatum* venom peptides. This led to the identification of a neuroactive peptide that paralyzed blowflies and honeybees when injected. This peptide, named U_11_-MYRTX-Tb1a (hereafter referred to as U_11_ throughout the manuscript), was shown to harbor a compact helix structure stabilized by a single disulfide bond, seemingly a unique three-dimensional scaffold among animal venoms. Preliminary pharmacological assays combined with structural features converge toward the hypothesis that U_11_ may modulate potassium channels.

## 2. Results

### 2.1. Neuroactivity Screen of Venom Peptides and Selection of U_11_

To identify novel insecticidal peptides from the venom of the ant *T. bicarinatum*, we screened nine synthetic peptides by intrathoracic injection of a high dose of each peptide into the blowfly *Lucilia caesar* (Table 1).

At the doses tested, four peptides (U_8_, U_10_, U_11_, and U_13_) paralyzed flies 1 h after injection, of which only U_11_ caused paralysis of all flies (ten flies per replicate) in all three replicates, yet at the lowest dose tested among the peptides (i.e., 27 nmol. g^−1^; doses are normalized to the average body weight of the injected prey for further comparison among different insect species) (Figure 1A). Based on this result, we selected U_11_ for further investigation and determined its potency to paralyze flies 1 h and 24 h after injection. These experiments confirmed that U_11_ is a potent neurotoxic peptide capable of inducing rapid flaccid paralysis in flies with a PD_50_ value of 1.78 ± 0.12 nmol. g^−1^ 1 h post injection. The paralysis was irreversible 24 h post injection with a PD_50_ value of 2.67 ± 0.13 nmol. g^−1^ (Figure 1B). We then extended our experiments by injecting U_11_ into honeybees (*Apis mellifera*) and aphids (*Acyrthosiphon pisum*) to evaluate whether the peptide was active against beneficial and pest insects. U_11_ was significantly more capable of paralyzing honeybees than blowflies 1 h post injection (the PD_50_ value was 1.36 ± 0.08 nmol. g^−1^) and 24 h post injection (PD_50_ value was 1.11 ± 0.09 nmol. g^−1^) (Figure 1B). In contrast, U_11_ had only a weak effect when injected into *A. pisum* at a dose of 3.4 nmol. g^−1^ (6.6% of aphids were affected 24 h post injection) (Figure 1B) and on aphid fitness when monitored over 7 days (Appendix A).

Based on these injection assays alone, U_11_ does not appear to be relevant for agronomic development due to its neurotoxic effects on bees and its weak effect on aphids. However, although most venoms are injected substances, they may contain some insecticidal peptides that are active when ingested or by contact on pests, yet are safe for beneficial insects [25]. Such a feature is a key point to turn an insecticidal venom peptide into a commercialized bio-insecticide. We therefore pursued the investigation of U_11_ insecticidal activities by testing its toxicity by ingestion at high doses on a panel of both model (*L. caesar*, *D. melanogaster*, *A. mellifera*) and pest insects (*A. pisum* and *Sitophilus oryzae*). U_11_ was active by ingestion against both dipteran species (Appendix A), while honeybees, aphids, and weevils remained unaffected (Appendix A). However, the flies were only affected at high doses. For instance, 60% of blowflies were affected 24 h post ingestion with a dose of 8 mg/mL (i.e., 350 nmol. g^−1^), which is 130-fold higher than the PD_50_ value when injected. This illustrates that oral toxicity can alter the susceptibility of some insects such as pollinators and must be considered in future biodiscovery programs for new insecticides.

### 2.2. Mechanism of Action of U_11_

Since most of the reported ant venom peptides that exhibit insecticidal activities are cytotoxic [21,26,27,28,29] (see Table 2), we tested U_11_ cytotoxicity using in vitro assays on the dipteran S2 *Drosophila* embryonic cell line. Both CCK-8 and LDH assays showed that U_11_ was not cytotoxic to S2 cells after 1 h and 24 h of incubation at 5 and 50 µM (Figure 2A,B).

Altogether, the aforementioned results indicate that U_11_ is neurotoxic when injected into dipterans and bees, without affecting the cell membrane. We therefore hypothesized that ion channel modulation may be responsible for the observed neuroactivity of U_11_. To investigate the cellular mechanisms by which U_11_ causes paralysis in insects, we first examined whether U_11_ modulated the membrane potential of S2 cells using the voltage-sensitive fluorescent molecular probe DiBAC_4_(3). Incubation of S2 cells with 10 µM U_11_ significantly decreased the KCl depolarization (Figure 2C). A depolarization-induced increase in fluorescence intensity in the presence of U_11_ is 33% less than in the control condition (calculated AUC values are 312 ± 23 and 464 ± 35, respectively). We then tested whether the U_11_ peptide was able to elicit Ca^2+^ mobilization by using the intracellular Ca^2+^ indicator Fluo-4/AM probe. No effect on calcium mobilization in the intracellular Ca^2+^ stock (Tg response) or in the calcium influx (Figure 2D) was observed after incubation of S2 cells with 10 µM U_11_.

### 2.3. NMR Structure of U_11_

Based on the primary structure of U_11_, which has no similarity to other venom toxins except for another U_11_ peptide (i.e., U_11_-MYRTX-Ta1a) described in the venom of *T. africanum* [30], we considered that the potential to discover an original 3D structure was high and that it might provide insights into U_11_‘s mode of action. The 3D structure of U_11_ was therefore determined by using an NMR spectroscopy. The 1D ^1^H-NMR (Appendix A) and ^15^N-HSQC (Appendix A) spectra of the peptide revealed a good dispersion of the amide proton and nitrogen chemical shifts, indicative of a well-folded peptide. Following the standard assignment strategy, analyses of the 2D-TOCSY and NOESY spectra allowed an almost complete assignment of the observable ^1^H chemical shifts (99.6–96% of all protons). The natural abundance ^13^C-HSQC NMR spectrum (Appendix A) helped us to unambiguously assign the ^1^H chemical shifts, particularly in crowded regions of the ^1^H TOCSY and NOESY spectra corresponding to residue side chains. The 3D structures were calculated by considering a total of 738 NOE-derived distance restraints, 14 hydrogen bonds, and 48 dihedral angles (Appendix A). Among the 500 water-refined structures of U_11_, the 15 structures of lowest total energy in agreement with all the experimental data and the standard covalent geometry were used for the statistical analysis (Appendix A). The analysis of these 15 final structures with PROCHECK-NMR [31] showed that 89% of the residues are in the most favored or additionally allowed regions of the Ramachandran plot (Appendix A).

The solution structure of U_11_ is presented in Figure 3. The N-terminal part of the molecule appears as a series of non-canonical β-turns (type IV) extending from residues Gly1 to Lys6, followed by a short turn of 3_10_-helix from Leu7 to Gln9 (Figure 3A,B). After that, residues Cys10 to Cys33 delineate an almost planar triangular monocycle closed by the disulfide bridge (Figure 3B). This ring encompasses two helical domains: the first one is an α-helix (Ala17 to His24), while the second is a short 3_10_-helix (Glu27 to Leu30). The inter-helical angle determined by PROMOTIF is (120.7 ± 2.3); the two helices define a planar equilateral triangular monocycle closed by the disulfide bridge. Besides the disulfide bond (Cys10-Cys33), several factors contribute to the compactness of the structure: (i) hydrogen bonds form between the Cys33 and Ile34 backbone amides and the Glu9 backbone carbonyl; (ii) hydrophobic interactions between the aliphatic part of the side chain of Lys6, Leu7, Phe11, Met14, Ile19, Ala22, and Phe31, which form the peptide’s hydrophobic core, inducing folding of helix 1 on the ring; (iii) ionic interactions between Lys8 ε-ammonium group and both Glu3 γ-carboxyl group (8 structures out of 15) and/or Ile34 carboxyl-terminus (8 structures out of 15) position the N-terminus closer to the ring. U_11_ is a highly charged molecule (Figure 3C) featuring nine lysine, three glutamate, and two aspartate residues. All charged residue side chains are exposed to the solvent and distributed around the molecular surface, especially on the side of the ring that includes the N-terminus (Figure 3D, right panel). On the other side (Figure 3D, left panel), U_11_ presents a small groove bordered by the hydrophobic residues Leu16, Ala17, Ala18, and Leu30.

## 3. Discussion

A recent study has revealed the peptide repertoire contained in the venom of *T. bicarinatum* [20]. Most of these peptides were novel sequences that bore no sequence similarity to other reported venom peptides and whose biological activities remained to be elucidated. In this paper, we focused on the potential insecticidal activity of some of these peptides. A paralytic screen performed on the dipteran *L. caesar* with nine *T. bicarinatum* venom peptides of unknown biological activity led to the selection and further investigation of U_11_, a 34-amino acid peptide. The determination of the U_11_ structure revealed a compact helix ring peptide, which is stabilized by a single disulfide bond and other intramolecular interactions. The three helices are closely packed and form a highly charged globular structure. The venoms of wasps, bees, and ants are dominated by linear peptides, but toxins structured by one disulfide bond have been occasionally noted in venoms from hymenopterans (e.g., secapin) [10,32]. The full length precursor sequence of *T. bicarinatum* U_11_ shares 68% sequence identity (76% sequence similarity) with the precursor peptide U_11_-MYRTX-Ta1a (GenBank accession number: OW518837.1) from *T. africanum* venom (Appendix A) [30], while identity reaches 74% (85% sequence similarity) for the mature region. There is no significant sequence similarity with other reported venom peptides, including in ants. The helical structure of U_11_ is somewhat evocating of the helical arthropod-neuropeptide-derived (HAND) toxins from spider and centipede venoms [33]. However, the number and type of helices, the number of cysteines, and the overall scaffold are very different. We further searched for structural neighbors in the Protein Data Bank using DALI [34], which yielded no match with the U_11_ structure. U_11_ appears therefore to define a new scaffold in animal venom peptides.

Future ant venom studies should investigate whether U_11_ peptides are present in other ant lineages or are restricted to the *Tetramorium* genus. To date, U_11_ appears to be the third most potent neuroactive ant venom peptide reported against the blowfly (Table 2). Such a potent neurotoxin may have contributed to the evolutionary success of this genus (587 valid species belong to the genus *Tetramorium* [5]) by enhancing the ability of ant workers to incapacitate arthropod prey. When injected, U_11_ caused paralysis in blowflies and honeybees but had only a weak paralytic effect when intra-abdominally injected into the aphid *A. pisum* at a dose equivalent to the PD_50_ values for blowflies and honeybees (3.4 nmol. g^−1^). Thus, either the aphids are resistant to the paralytic effect of U_11_, or the aphid’s morphology makes them less sensitive to injection than flies and bees. The lack of a clear separation between the thorax and abdomen in aphids may indeed dilute the toxin with the hemolymph of the whole body, rather than just the thoracic compartment as in flies and bees. It would be interesting in a future study to observe the effect of U_11_ at higher doses in aphids. Regarding resistance, ants, including those of the genus *Tetramorium,* are known to tend aphids in mutualistic interactions [35]. Therefore, selective pressures may have favored the emergence of ant venom resistance in aphids. A few or even a single amino acid variation in a pharmacological receptor may be sufficient to disrupt the affinity of a toxin for that receptor, rendering some organisms less sensitive to the toxin [36]. Identification of the pharmacological target of the U_11_ toxin would enable us to examine whether this receptor is conserved across insect lineages and would provide molecular insights into the differences observed in our paralytic assays.

Most insecticidal peptides isolated from ant venoms so far are cytotoxic, disrupt cell membranes, and are multifunctional, including antimicrobial, algesic, and insect neurotoxic activities [27]. These membrane-active peptides act by triggering an influx of extracellular Ca^2+^ through membrane cell perturbation [17]. In this study, we demonstrated that U_11_ is not cytotoxic and does not induce Ca^2+^ mobilization, but can modulate a KCl-elicited membrane depolarization on the dipteran S2 cells. Given that the membrane potential recordings were conducted in the presence of the non-permeant cation N-Methyl-D-glucamine (NMDG) instead of extracellular Na^+^, the observed depolarization was not caused by a Na^+^ influx. This absence of effect on Ca^2+^ signaling, coupled with the Na^+^-independent depolarization, suggests that the U_11_ mechanism of action involves K^+^ conductance. Extensive electrophysiological experiments with a variety of selective ion channel inhibitors would be necessary to confirm these results and to gain a comprehensive understanding of U_11_ pharmacology.

Potassium channels are transmembrane proteins that regulate the flow of potassium ions across the cell membrane, playing a crucial role in regulating the electrical activity of cells. Given their potential application for therapeutic purposes, potassium channel ligands have been extensively studied. To date, 422 natural polypeptide ligands of potassium channels have been curated in the Kalium database [37]. Only twenty of them originate from non-venomous organisms, while the rest come from venomous animals (i.e., 212 from scorpions, 66 from spiders, 44 from sea anemones, 35 from snakes, 36 from cone snails, and the remaining 19 from other venomous arthropods or from a venomous lizard). Apamin, a selective inhibitor of K_Ca_2 channels [38], and tertiapin, which blocks the inwardly rectifying K_ir_1.1 and K_ir_3.1/3.4 channels [39], are the only two Hymenoptera venom peptides described to target potassium channels. Of different origins, all these natural toxins display different sizes, primary sequences, cysteine connectivity, and exhibit a variety of folding [40,41]. It is noteworthy that, among these, only two venom peptides, yet active on various types of potassium channels, present a single disulfide bridge. One is the AnmTX BC 9a-3, part of a group of three homologous toxins from the sea anemone *Bunodosoma cangicum* that share conservative amino acids but whose cysteine numbers and positions differ [42]. The other single-bridge toxin is contryphan-Vn from *Conus ventricosus*, an atypical nine residue peptide containing a D-tryptophan that affects both voltage-gated and calcium-dependent potassium channel activities [43]. Therefore, if activity from U_11_ on potassium channels is confirmed, it would be the third with a single disulfide bridge. At least eight different folds for K^+^ channel-active toxins have been identified, ranging from all-sheet peptides (e.g., marine cone snail κ-PVIIA or spider hanatoxin) to all-helical peptides (e.g., scorpion hefutoxin or sea anemone ShK and BgK) and combinations of both (e.g., snake dendrotoxin or scorpion charybdotoxin). From the current work, the U_11_ structure belongs to a 3_10_α3_10_ type of fold. Despite their great structural diversity, potassium channel modulators share some common features and most of them are basic toxins with a pI > 7 [44], as is U_11_ with a pI of 9.3.

Toxins can affect voltage-gated potassium channels primarily through two distinct targeting mechanisms. Toxins called gating modifiers interact with defined regions within the voltage sensor, thereby affecting the gating motions and modifying the energy of the voltage-dependent gating process [45,46,47,48]. Others, the majority of which are known as pore blockers, bind to the outer vestibule on the N-terminal side of the re-entrant pore loop, spanning between the S5 and S6 transmembrane domains. To inhibit the channel function by physically occluding the potassium permeation pathway, many structurally unrelated potassium pore-blocker peptides share two pivotal residues, referred to as the ‘functional dyad’ [48,49,50,51]. These dyads consist of a basic lysine, which in the toxin–channel complexes has been shown to protrude into the selectivity filter [52], and a hydrophobic aromatic amino acid (usually tyrosine or phenylalanine), whose contacts with a cluster of aromatic channel residues, together with other electrostatic contacts, stabilize the toxin–channel interaction. An examination of the U_11_ structure revealed two potential dyads: Lys2-Phe11 or Lys25-Tyr21 (Figure 4A). Given that Phe11 is not conserved in the *T. africanum* U_11_ homolog and Lys2 lies in a flexible region of the peptide, the Lys25-Tyr21 pair appears more likely as a putative pharmacophore. It is present on a flat surface of the peptide, as it is in the sea anemone peptide BgK (Figure 4B), and the distances between the residues are 7.2 ± 0.2 Å. Although not all pore blockers exhibit such dyads (for examples, see HelaTx1 [52] or Tc32 [53]), when present, they spatially superimpose whatever the toxins folds [46,47] (Figure 4C). The above-mentioned structural features reinforce the preliminary pharmacological results, supporting the potassium channel inhibitory activity of U_11_. However, it should be noted that the sea anemone venom peptide OspTx2b, which is homologous to BgK and ShK and contains the Lys-Tyr dyad, has been shown to lack K_V_ channel blocking activity [54]. Further electrophysiological experiments are therefore required to confirm the current hypotheses.

## 4. Conclusions

Ant venoms hold great promise but remain underexplored compared to other venomous organisms such as arachnids, snakes, and cone snails. Furthermore, very few studies have been undertaken to characterize the insecticidal activity of ant venom toxins. This paper describes the unveiling of the U_11_ toxin, an original neuroactive peptide from *T. bicarinatum* ant venom that causes irreversible paralysis in blowflies and honeybees. Our pharmacological and structural data converge toward the possible modulation of potassium channels, a highly valuable molecular target for the development of new drugs. Future in-depth electrophysiological investigations should provide a wealth of information to decipher the mode of action and to expand the description of U_11_ pharmacology. The findings presented in this work have already revealed the promising biological properties of U_11_, further emphasizing the potential of ant venoms as an untapped resource of ion channel ligands, along with novel toxin scaffolds that could be exploited as drugs or insecticides leads.

## 5. Materials and Methods

### 5.1. Peptide Synthesis

Chemical reagents. All Fmoc amino acid residues, O-benzotriazol-1-yl-N,N,N′,N′-tetramethyluronium hexafluorophosphate (HBTU), and Rink amide 4-methylbenzhydrylamine (MBHA) resin were purchased from Novabiochem (Fontenay sous Bois, France) or IRIS Biotech (Marktredwitz, Germany). Preloaded 4-hydroxymethyl-phenoxymethyl-copolystyrene-1%-divinylbenzene resins (HMP) were obtained from Life Technologies (Villebon sur Yvette, France). N,N-Diisopropylethylamine (DIEA), piperidine, trifluoroacetic acid (TFA), triisopropylsilane (TIS), tert-butylmethylether (TBME), dimethylsulfoxide (DMSO) were supplied from Sigma-Aldrich (Saint-Quentin-Fallavier, France). N-methylpyrrolidone (NMP), dimethylformamide (DMF), dichloromethane (DCM), acetonitrile and acetic acid were from Fisher Scientific (Illkirch, France).

A panel of nine synthetic peptides characterized from the venom of *T. bicarinatum* (U_2_, U_4_, U_7_, U_8_, U_10_, U_11_, U_13_, U_14_, and U_15_), used for the insecticidal screening and insecticidal assays on honeybee and NMR experiments, were synthesized by Fmoc solid phase methodology on a Liberty microwave-assisted automated peptide synthesizer (CEM, Saclay, France) using the standard manufacturer’s procedures at 0.1 mmol scale. All Fmoc amino acids (0.5 mmol, 5 eq.) were coupled on preloaded HMP resin or Rink amide resin, by in situ activation with HBTU (0.5 mmol, 5 eq.) and DIEA (1 mmol, 10 eq.) before Fmoc removal with 20% piperidine in DMF. After completion of the chain assembly, peptides were deprotected and cleaved from the resin by adding 10 mL of an ice-cold mixture of TFA/TIS/H_2_O (9.5:0.25:0.25, *v*/*v*/*v*) and agitating for 3 h at room temperature. After precipitation in TBME followed by centrifugation (4500 rpm, 15 min), the crude linear peptides were purified by reversed-phase HPLC (Gilson, Villiers le Bel, France) on a 21.2 × 250 mm Jupiter C_18_ (5 µm, 300 Å) column (Phenomenex, Le Pecq, France) using a linear gradient (10–40% for U_4_, U_7_, U_14_; 10–50% for U_2_; 10–60% for U_10_, U_15_; 20–50% for U_11_; 20–60% for U_8_ and 30–60% for U_13_ over 45 min) of acetonitrile/TFA (99.9:0.1) at a flow rate of 10 mL/min. The formation of the disulfide bond for peptides U_4_, U_7_, and U_11_ was then carried out by oxidation of purified linear peptides in a mixture of H_2_O/CH_3_COOH/DMSO (75:5:20) (0.5 mg/mL) at pH 6. The reactions were stopped after 20 h, freeze-dried, and the resulting cyclic peptides were purified in the same conditions as described above. Peptides were then characterized by MALDI-TOF mass spectrometry on a ultrafleXtreme (Bruker, Strasbourg, France) in the reflector mode using α-cyano-4-hydroxycinnamic acid as a matrix. Analytical RP-HPLC, performed on a 4.6 × 250 mm Jupiter C_18_ (5 µm, 300 Å) column, indicated that the purity of the peptides was >99.9%.

### 5.2. Injection Assays

Insecticidal assays with venom peptides are usually conducted against a single insect model that does not reflect the genuine functional role of the toxins as well as their potential for bioinsecticide development. In this study, we screened the neurotoxic activity of U_11_ when injected into a panel of beneficial and pest insects such as honeybees, blowflies, and aphids. Blowfly is a dipteran model commonly used to evaluate the insecticidal potency of venom peptides [25]. Blowfly maggots (*Lucilia caesar*) were bought from a fisheries shop (Euroloisir81, Lescure-d’Albigeois, France) and were kept at 25 °C until hatching. Flies 1–4 days post hatching were used for the injection assays. Honeybee (*Apis mellifera*) is a model of choice to assess the impact of a toxin against beneficial insects such as pollinators. The honeybee colonies used in this study belong to the Buckfast breed and were maintained at the University Champollion campus in Albi. Adult foraging honeybees were individually isolated in plastic vials and immediately injected. Assays on blowflies and honeybees were conducted by lateral intrathoracic injection of peptides dissolved in ultra-pure water at several concentrations (from 0.005 to 0.52 mM for *L. caesar* and from 0.002 to 1 mM for *A. mellifera*) by using a 1.0 mL Hamilton Syringe (1000 Series Gastight, Hamilton Company, Reno, NV, USA) with a fixed 25 gauge needle attached to an Arnold hand microapplicator (Burkard Manufacturing Co., Ltd., Rickmansworth, UK). Each fly/bee received 1 µL of peptide solution and was individually housed in a 2 mL tube with 3 µL of 5% glucose solution. Paralytic activity was monitored at 1 h and 24 h after injection while lethality was monitored only at 24 h. Flies/bees that showed no sign of paralysis (no sign of movement dysfunction) were categorized as unaffected; otherwise, they were noted as paralyzed. Flies/bees were considered dead if they did not react at all after mechanical stimulations with tweezers monitored under dissecting microscope. Ten flies/honeybees were used for each toxicity experiment and for the appropriate control (ultrapure water solution), and the experiment was repeated three times for each dose. Dose–response data were analyzed as detailed by Touchard et al. [15]. The pea aphids *A. pisum* used in this study were obtained from a long-established parthenogenetic clone (LL01). Aphids were kept on young broad bean plants (*Vicia faba,* L. cv. Aquadulce) at 21 °C with a photoperiod of 16 h light 8 h dark, allowing us to maintain them as strictly parthenogenetic matrilines. To produce synchronized apterous nymphs, winged adults were allowed to reproduce on seedlings for 24 h. The resulting synchronized nymphs were reared for 9 days on plants before the beginning of the experiments (peptide injection or feeding). For the injection experiment, we used four groups of synchronized A9 aphids (30 per group): (i) non-injected aphids, (ii) aphids injected with sterilized ultra-pure water (injection control), (iii) aphids injected with PA1b inactivated peptide (non-toxic peptide injection control) [55], and (iv) aphids injected with the U_11_ peptide. The injection protocol was the one developed by Sapountzis et al. [56]. Briefly, to minimize the mortality associated with microinjections, a volume of 46 nL was injected between the 2nd and the 3rd abdominal segment using an automatic injector apparatus Nanoject III (Drummond Scientific, Broomall, PA, USA) with 1.0 mm O.D. × 0.78 mm I.D. capillaries. Aphids were immobilized with a homemade vacuum-operated insect holder for accurate positioning for intra-abdominal injections. PA1b-inactivated and U_11_ peptides were administered at the concentration of 1 mg/mL. All injected aphids were directly placed back to plants after injections. Aphids were monitored daily for 7 days after injection and their survival and behavior regularly checked, following the protocol developed by Ribeiro Lopes et al. [57]. Every 3 days, aphids were moved to a new plant to keep only the treated ones and eliminate newborn nymphs. The average weight of flies (19 ± 2 mg), bees (109 ± 9 mg), and aphids (3.4 ± 0.7 mg) was used for the comparison of doses according to the type of insects injected.

### 5.3. Ingestion Assays

We used the same protocol used in Guo et al. [25] for the ingestion assays with flies (*L. caesar* and *D. melanogaster*). Briefly, flies were placed in 2 mL microcentrifuge tubes right after hatching and were individually fed with 3 µL of 50% *w*/*v* sucrose solution (negative control) or sucrose solution containing the toxin. They were then kept at 25 °C. Mortality was recorded at 1, 2, 4, 24, 48, and 72 h after exposure to the tested compound. The surviving flies received 3 µL of 50% *w*/*v* sucrose solution without peptide each day until the end of the experiment. For each test, ten flies were used per treatment and each of them was replicated three times.

For the feeding experiments on aphids, one-day-old aphids (corresponding to the first-instar nymphs N1) were placed for seven days in feeding chambers on 48 μL of a nutritive artificial diet specifically developed for *A. pisum* feeding, and development was completed [58] with or without the peptide U_11_. The diets were changed after three days to avoid contamination. Ten N1 were deposited in the feeding chamber and three replicates were performed for each bioassay. Aphids’ mortality was assessed every day and compared between the different diets using a GLM linear regression model after observation of the residuals with a significance level of 5% on the R Studio software. Survival of aphids on two diets, completed, respectively, with U_11_ at 500 μg/mL and 1 mg/mL, was followed during seven days.

*S. oryzae* toxicity tests were performed with the WAA42 strain as described previously [58]. Briefly, the insects used for the bioassays were adults 2 to 3 weeks of age collected in one week experimental cohorts and placed in batches of 30 individuals on feed pellets containing the synthetic U_11_ tested in a whole grain-based diet. The tested concentration of the U_11_ was 800 μg/g of the feed. Bioactivity was assessed by daily insect survival during the first two weeks of contact with the test diet (27.5 °C, 70% room humidity) and by standard survival analysis.

The ingestion assays on honeybees were conducted by placing bees in 850 mL plastic storage boxes with 2 mL microcentrifuge tubes filled with 50% *w*/*v* sucrose solution and previously drilled for bee access. The boxes were kept at 25 °C in the dark and the bees were allowed to feed ad libitum on the sucrose solution. For each test, ten bees were used per treatment, and each of them was replicated three times. Lethality was recorded at 1, 2, 4, 24, and 48 h after exposure to U_11_ peptide.

For oral toxicity assays, insects were starved for 2 h prior testing. Bees were individually force fed with 5 µL of 50% *w*/*v* sucrose solution (negative control), or sucrose solution containing the toxin (8 mg/mL–110 nmol/g bee).

### 5.4. S2 Cell Culture

The embryonic S2 *Drosophila* cell line (Thermofisher, Waltham, MA, USA) was maintained in Schneider’s insect medium (Sigma-Aldrich, Saint Louis, MO, USA) complemented with 10% heat-inactivated fetal bovine serum (FBS) (Sigma-Aldrich) and 1% Penicillin/Streptomycin (Sigma-Aldrich) at 25 °C without CO_2_ on TC-treated support (Corning^®^, Glendale, AZ, USA). The FBS was heat inactivated at 60 °C for 45 min. The cells were passaged every two days at about 90% confluence.

### 5.5. Cytotoxicity Assays

The CCK-8 (BosterBio, Pleasanton, CA, USA) and LDH (Dojindo, Rockville, MD, USA) assay kits were used to assess the cellular viability and mortality, with densities of 2 × 10^5^ cells/mL and 6 × 10^5^ cells/mL, respectively. Briefly, 100 µL of S2 cells at the appropriate densities were seeded in 96-well TC-treated plates and incubated for 24 h at 25 °C. Then, the peptide was added at a final concentration of 50 µM and 5 µM for 24 h at 25 °C. Lysis Buffer, Triton 0.1%, and culture medium were used for controls. Viability and mortality cell assays were realized separately and in triplicates according to manufacturer’s instructions. The absorbances were measured using a BioTek Cytation 1 microplate reader (Agilent, Santa Clara, CA, USA) (i.e., 450 nm and 490 nm for CCK-8 and LDH assays, respectively). The percentages of living or dead cells were then calculated according to manufacturer’s recommendations.

### 5.6. Pharmacological Experiments

Variations of membrane potential were assessed using the fluorescent dye bis-(1,3-dibutybarbituric acid) (DiBAC_4_(3)) (Thermofisher) using a IX73 fluorescence microscope (Olympus, Japan). Then, 6 × 10^5^ S2 cells were plated on 0.01% Poly-L-Lysine (Sigma-Aldrich) coated glass-bottom Petri dishes and allowed to adhere for 24 h before experiments. Cells were rinsed once with the assay buffer containing (in Mm) 115 N-Methyl-D-glucamine, 2 CaCl_2_, 1 MgCl_2_, 5 KCl, 48 Sucrose (Ph adjusted to 7.2 with KOH). Then, cells were loaded with 500 Nm DiBAC_4_(3) with or without the tested compound for 30 min at 25 °C in the assay buffer. Signal acquisition was performed at excitation and emission wavelengths of 494 nm and 518 nm, respectively, for 5 min with a 5 s sampling rate. Baseline was recorded for 1 min before addition of KCl (50 Mm final) to achieve membrane depolarization. The results are presented as transformed data to obtain the relative fluorescence to the time of addition of the stimulus, according to the equation: Fx = Ft/F0, where Ft represents the fluorescent value at a specific time point and F0 represents the mean basal fluorescent value. To compare the different conditions, areas under the curve (AUC) after KCl addition were calculated (between 60 and 300 s).

The [Ca^2+^]i mobilization was measured with the Fluo-4/AM probe using the imaging reader Cytation 1 (BioTek, Agilent). Then, 6 × 10^5^ S2 cells/mL were plated on glass-bottom Petri dishes and allowed to adhere for 24 h before experiments. Cells are incubated with 2 µM of Fluo-4/AM in cell culture medium with 2.5 mM probenecid for 1 h at 25 °C with 10 µM of U_11_ peptide. Then, cells were rinsed twice in Ca^2+^-free buffer containing (in mM) 120 NaCl, 5 KCl, 4 MgCl_2_, 32.2 Sucrose, 10 HEPES, 0.1 EGTA, 2.5 probenecide (pH 7.2), incubated in Ca^2+^-free buffer, and processed for measurement of cytoplasmic Ca^2+^ concentration. First, 100 µL of Thapsigargin (2 µM final) was injected and calcium images were acquired with FITC filter (λ_ex_: 485 nm/λ_em_: 528 nm) every 10 s for 5 min, then 100 µL of CaCl_2_ (1.8 mM final) was injected and images were acquired with FITC filter (λ_ex_: 485 nm/λ_em_: 528 nm) every 10 s for 7 min. Cell fluorescence was analyzed with CALIMA [59] (available on https://aethelraed.nl/calima, accessed on 3 April 2023). Changes in the Ca^2+^ cytosolic concentration were calculated from changes of fluorescence relative to time 0 (F = Ft/Ft0 (fluorescence at time = t/time = 0)).

### 5.7. NMR Experiments

U_11_ was solubilized in H_2_O/D_2_O (9:1 *v*/*v* ratio) to a final peptide concentration of 1.9 mM. pH was adjusted to 5.1. Then, 2D ^1^H-^1^H-TOCSY (Tm = 80 ms) [60], 2D ^1^H-^1^H-NOESY (Tm = 100 ms) [61], sofast-HMQC [62] (^15^N natural abundance), and ^13^C-HSQC [63,64] (^13^C natural abundance) spectra were recorded at 298K on an Avance III HD BRUKER 700 MHz spectrometer equipped with a cryoprobe. ^1^H chemical shifts were referenced to the water signal (4.77 ppm at 298K). The NMR data were processed using Bruker’s Topspin 3.2^TM^ and analyzed with CCPNMR (version 2.2.2) [65]. Structures were calculated using CNS [66,67] through the automatic assignment software ARIA2 (version 2.3) [68] with NOE-derived distances, hydrogen bonds (in accordance with the observation of the typical sequential and medium range NOE cross-peak network for α-helices—H^N^/H^N^, H^N^/H^α^, H^α^/H^β^), backbone dihedral angle restraints (determined with the DANGLE program [69]), and one imposed disulfide bond between Cys10 and Cys33. The ARIA2 protocol used simulated annealing with torsion angle and Cartesian space dynamics with the default parameters. The iterative process was repeated until the assignment of the NOE cross peaks was complete. The last run was performed with 2000 initial structures and 500 structures were refined in water. In total, 15 structures were selected based on total energies and restraint violation statistics to represent the structure of U_11_ in solution. The quality of the final structures was evaluated using PROCHECK-NMR and PROMOTIF [31,70]. The figures were prepared with PYMOL [71]. Electrostatic and hydrophobic potentials were determined at the Connoly surface of U_11_ using chimeraX 1.3 [72].

Appendix A have been provided.

## Figures and Tables

**Figure 1 toxins-15-00600-f001:**
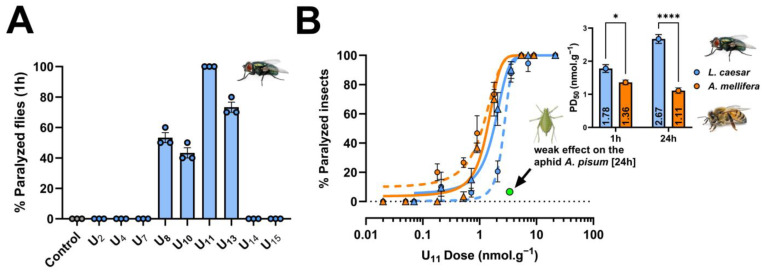
Neuroactivity of venom peptides by injection against three insect species. (**A**) paralytic screen of nine *T. bicarinatum* venom peptides against the *L. caesar* blowfly. Values (percentage of paralyzed flies 1h after injection) are expressed as mean ± SEM (*n* = 3 independent experiments indicated by circles). (**B**) dose–response curves for *L. caesar* blowflies (blue lines) and *A. mellifera* honeybees (orange lines) injected with U_11_, at 1 h (continuous lines and triangles) and 24 h (dotted lines and circles) after intrathoracic injection. Intra-abdominal injection of U_11_ into *A. pisum* aphids (green circle) showed weak paralytic effect at 3.4 nmol. g^−1^ (injection of 46 nL of a 0.25 mM solution of U_11_). Values are the percentage of paralyzed insects presented as mean ± SEM (*n* = 3 independent experiments) fitted with a non-linear regression with variable slope. Bar graph shows 50% paralytic dose (PD_50_) values; *, *p* < 0.05; ****, *p* < 0.0001 (two-way ANOVA followed by Tukey’s multiple comparison test).

**Figure 2 toxins-15-00600-f002:**
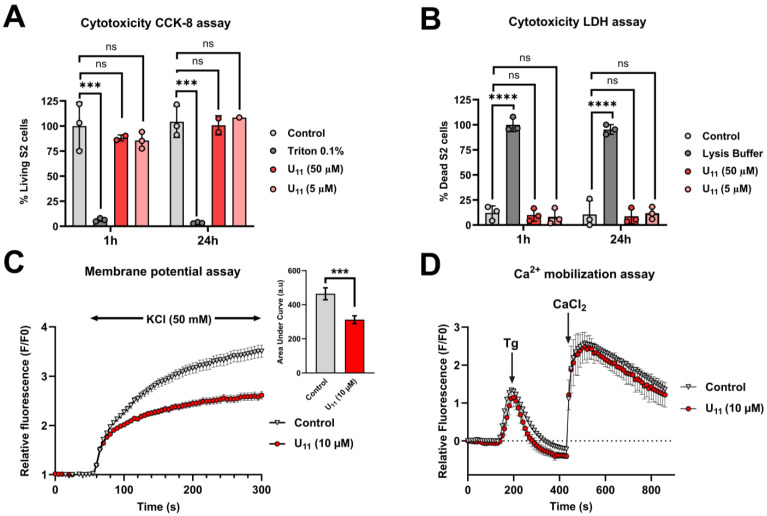
U_11_ pharmacological assays using *Drosophila* S2 cells. Cytotoxic activities of U_11_ against S2 cells obtained from CCK-8 (**A**) and LDH (**B**) assays. Values are represented as mean ± SEM (*n* = 1–3); ns, not significant; ***, *p* < 0.001; ****, *p* < 0.0001 (two-way ANOVA followed by Tukey’s multiple comparisons test). (**C**) membrane potential recorded by DiBAC_4_(3). Time course of S2 membrane potential after buffer (control) or 10 µM U_11_ incubation. Membrane depolarization, indicated by an increase in relative fluorescence, was induced by application of high KCl concentration (50 mM). Bar graph of corresponding area under curves (AUC). AUC were calculated after KCl application until the end of experiment. Values are represented as mean ± SEM of *n* = 220 cells (*n* = 3); ****, p* < 0.001 (Welch’s *t* test). (**D**) intracellular Ca^2+^ concentration was recorded with Fluo-4/AM probe in a Ca^2+^-free buffer after a preincubation of S2 cells for 1 h in cell culture medium without (control) or with 10 µM of U_11_. Increase in cytosolic Ca^2+^ concentration, indicated by an increase in relative fluorescence, was induced by injection of thapsigargin (Tg) at 2 µM (first peak) and CaCl_2_ at 1.8 mM (second peak). Values are represented as mean ± SEM of *n* = 442 cells (*n* = 6 Petri dishes).

**Figure 3 toxins-15-00600-f003:**
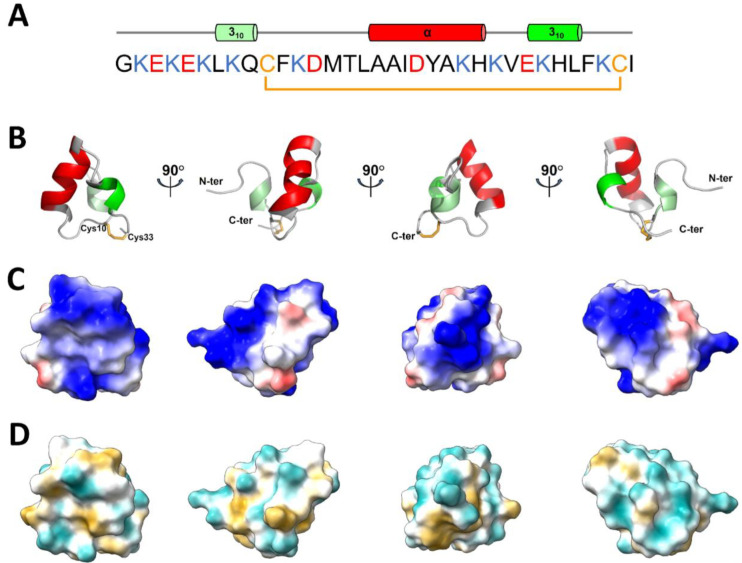
Three-dimensional structure of U_11_. (**A**) primary sequence of U_11_ with cationic, anionic and cysteine residues in blue, red, and orange, respectively. The secondary structural elements elucidated from the NMR experiments are shown above the sequence. (**B**) cartoon representation of the lowest-energy model representative of U_11_ structure in solution. Disulfide bridge 10–33 is displayed in orange. (**C**) coulombic electrostatic potentials (ESP) determined using chimeraX at the surface of U_11_. Surface coloring ranges from red for negative potential, through white, to blue for positive potential. (**D**) molecular lipophilic potentials (MLP) calculated at the surface of U_11_ using chimeraX. Surface coloring ranges from dark goldenrod for the most hydrophobic potentials, through white, to dark cyan for the most hydrophilic potentials.

**Figure 4 toxins-15-00600-f004:**
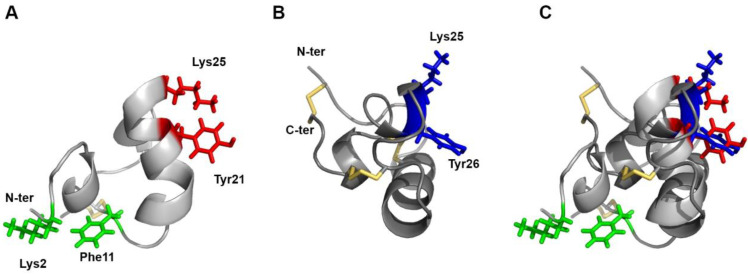
(**A**) ribbon representation of the structure of U_11_. The side chains of the two pairs of residues eligible as functional dyads are shown in red and green, respectively. (**B**) ribbon representation of the helical cross-like fold (CSα/α) of the sea anemone toxin BgK (PDB: 1BGK). Residues of the functional dyad are shown in blue. (**C**) superimposition of the functional dyad Lys25/Tyr21 (red) of U_11_, whose backbone is shown in light gray with the functional dyad Lys25/Tyr26 (blue) of BgK, whose backbone is in dark gray.

**Table 1 toxins-15-00600-t001:** Venom peptides from *T. bicarinatum* screened on blowfly (*L. caesar*) paralytic assay.

Peptide	Sequence	MW ^#^	Injected Dose ^¥^
U_2_	DPPPGFIGVR *	1052.6	103 nmol. g^−1^
U_4_	GCSQFRRMRNLCG *	1523.7	68 nmol. g^−1^
U_7_	AINCRRYPRHPKCRGVSA	2081.1	52 nmol. g^−1^
U_8_	GMLDRILGAVKGFMGS	1650.9	54 nmol. g^−1^
U_10_	GLGFLAKIMGKVGMRMIKKLVPEAAKVAVDQLSQQQ	3882.2	28 nmol. g^−1^
U_11_	GKEKEKLKQCFKDMTLAAIDYAKHKVEKHLFKCI	4018.1	27 nmol. g^−1^
U_13_	RPPQIGIFDQIDKGMAAFMDLFK *	2636.4	45 nmol. g^−1^
U_14_	IPPNAVKSLQ *	1064.6	95 nmol. g^−1^
U_15_	VFLTPDQIKAMIKRH *	1795.0	48 nmol. g^−1^

“*” C-terminal amidation, ^#^ Molecular weight in g.mol^−1^, ^¥^ body weight average of blowfly was used to calculate the doses, “C” Cysteines involved in disulfide bridges.

**Table 2 toxins-15-00600-t002:** Neuroactivity of ant venom peptides on blowfly species (*Lucilia* spp.) ranked by PD_50_ values 1 h post injection.

Peptide	Venom	Blowfly Species	PD_50_ [1 h] *	Cytotoxicity	Reference
Rm1a	*Rhytidoponera metallica*	*L. caesar*	0.3 ± 0.0	N.D.	[23]
Nc3a	*Neoponera commutata*	*L. cuprina*	0.5 ± 0.0	Yes	[27]
U_11_-Tb1a	*Tetramorium bicarinatum*	*L. caesar*	1.8 ± 0.1	No	This study
Pp1a	*Pseudomyrmex penetrator*	*L. cuprina*	2.4 ± 0.6	Yes	[28]
U_20_-Mri1a	*Manica rubida*	*L. caesar*	2.9 ± 0.9	N.D.	[15]
Ta2a	*Tetramorium africanum*	*L. caesar*	7.1 ± 0.9	N.D.	[23]
Ae1a	*Anochetus emarginatus*	*L. cuprina*	8.9 ± 3.1	N.D.	[10]
U_10_-Mri1c	*Manica rubida*	*L. caesar*	10.5 ± 1.7	N.D.	[15]
M-Tb1a	*Tetramorium bicarinatum*	*L. caesar*	12.0 ± 1.0	Yes	[21]
U_10_-Mri1a	*Manica rubida*	*L. caesar*	12.1 ± 1.5	N.D.	[15]
U_9_-Tb1a	*Tetramorium bicarinatum*	*L. caesar*	13.1 ± 3.0	Yes	[21]
Na1b	*Neoponera apicalis*	*L. cuprina*	25.8 ± 13.9	Yes	[27]
Nc3b	*Neoponera commutata*	*L. cuprina*	26.4 ± 5.9	Yes	[27]
Nc1a	*Neoponera commutata*	*L. cuprina*	31.4 ± 6.3	Yes	[27]
Nc2a	*Neoponera commutata*	*L. cuprina*	38.1 ± 18.5	Yes	[27]
U_10_-Mri1b	*Manica rubida*	*L. caesar*	58.4 ± 7.9	N.D.	[15]
U_13_-Mri1a	*Manica rubida*	*L. caesar*	69.2 ± 8.6	N.D.	[15]
Pc1a	*Paraponera clavata*	*L. caesar*	74.3 ± 2.1	N.D.	[23]
Ta3a	*Tetramorium africanum*	*L. caesar*	77.8 ± 1.4	N.D.	[23]

* Values are expressed in nmol. g^−1^; N.D. = no data.

## Data Availability

Atomic coordinates for the final ensemble of 15 U_11_ structures were deposited in both the Protein Data Bank (PDB ID 8PWT) and Biological Magnetic Resonance Bank (BMRB ID 34837) databases.

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
