# Peer review of "Discovery of an Insect Neuroactive Helix Ring Peptide from Ant Venom"

_toxins, 2023, doi:10.3390/toxins15100600_

Round 1
Reviewer 1 Report
The manuscript describes the characterization of ant (Tetramorium bicarinatum) venom peptide U11.
Comments:
1) Show all the NMR spectra in the Supplementary Information.
2) Organism names should be in italics
3) "... venomous organisms [16], including several ant 75 species [17, 18]... (cite https://doi.org/10.1073/pnas.2112630119)
4) It is unclear how an increased peptide dose has no effect, but a lower dose has. More discussion with references is needed.
5) The NMR methods section has not mentioned using 2D COSY or low-mixing TOCSY. So, how the stereo-specific assignments were made for the residues like Ile, Thr ... in the U11 sequence?
6) Are there any low-intensity peaks in NMR spectra? All the data needs to be shown in the SI file.
7) Authors should cite the papers for TOCSY, NOESY 13C-HSQQC NMR experiments.
8) Should also discuss the other venom peptides/analogs that contain the Lys-Tyr diad (OspTx2b (https://doi.org/10.1016/j.toxicon.2018.05.006), and AsK132958 (https://doi.org/10.1016/j.peptides.2017.10.001)) but have NO Kv-channel blocking activity.
Author Response
The manuscript describes the characterization of ant (Tetramorium bicarinatum) venom peptide U11.
Comments:
1) Show all the NMR spectra in the Supplementary Information.
As suggested by Reviewer 1, figures S4, S5 and S6 of the homonuclear and natural abundance heteronuclear NMR spectra have been added to the Supplementary Information.
2) Organism names should be in italics
Totally agree. We missed some italics. This has been fixed.
3) "... venomous organisms [16], including several ant species [17, 18]... (cite https://doi.org/10.1073/pnas.2112630119).
Thanks for the suggestion, but it is not so appropriate to support our statement. This very interesting paper concerns the study of the peptide MIITX2-Mg1a from M. gulosa venom. To justify the statement, it is more relevant to cite the work that has to do with proteo-transcriptomic characterization of M. gulosa venom, already cited here (Robinson et al, Sci. Adv. 2018, 4, eaau4640).
4) It is unclear how an increased peptide dose has no effect, but a lower dose has. More discussion with references is needed.
This is not the case in our data. We think there may have been some confusion in Figure 1B between the 1h and 24h data, as they were both represented by circles. Thus, Figure 1B has been modified for greater clarity by adding triangles for the 1h paralysis data. If that doesn’t answer the question, could Reviewer 1 clarify the point he raised.
5) The NMR methods section has not mentioned using 2D COSY or low-mixing TOCSY. So, how the stereo-specific assignments were made for the residues like Ile, Thr ... in the U11 sequence?
If Reviewer 1's question is about how we proceeded to unambiguously assign the side chain protons, we've made this clear in the results section, mentioning that “the natural-abundance heteronuclear 13C-HSQC NMR spectrum helped us to unambiguously assign the 1H chemical shifts, particularly in crowded regions of the 1H TOCSY and NOESY spectra corresponding to residue side chains”. Indeed, to avoid any ambiguity, the sentence "The natural-abundance heteronuclear NMR spectra helped us to unambiguously assign the 1H chemical shifts" (lines 194-195) has been changed to "The natural-abundance heteronuclear 13C-HSQC NMR spectrum helped us to unambiguously assign the 1H chemical shifts", since the 15N-HSQC spectrum is not useful for proton assignment. Figure S6 of the 13-HSQC spectrum added to the Supplementary Information upon Reviewer 1 recommendations illustrates this point.
6) Are there any low-intensity peaks in NMR spectra? All the data needs to be shown in the SI file.
As can be seen from the NMR spectra on Figures S4, S5 and S6, there are no split or low-intensity peaks that would indicate the presence of a minority form of the U11 peptide. Otherwise, it would have been mentioned in the text.
7) Authors should cite the papers for TOCSY, NOESY 13C-HSQQC NMR experiments.
Though usually not cited any more, the requested references have been added.
8) Should also discuss the other venom peptides/analogs that contain the Lys-Tyr diad (OspTx2b (https://doi.org/10.1016/j.toxicon.2018.05.006), and AsK132958 (https://doi.org/10.1016/j.peptides.2017.10.001)) but have NO Kv-channel blocking activity.
It is indeed a very interesting counter-argument that needs to be considered in our discussion. We have added two sentences about the OspTx2b toxin in discussion lines 339-343 as well as the suggested reference.
However, we have not discussed so much the reference to the toxin AsK132958. The article states that this toxin has no effect on potassium channels likely because it does not possess the functional dyad, contrary to what the reviewer says.
Reviewer 2 Report
It is a well-written paper that describes a novel structural family of neuroactive venom peptides from Tetramorium bicarinatum, one of the most successful and widespread tramp ant species. This is a significant discovery. The experiments were well-designed, the data were well-interpreted, and the conclusion is supported by the data.
The injected U11 caused paralysis in blowflies and honeybees but not in aphid A. pisum. The authors speculated that either the aphids are resistant to the paralytic effect of U11, or the aphids' morphology makes them less sensitive to injection than flies and bees. As the authors stated, the selectivity of ant venom on the pharmacological receptors of their prey may have been fine-tuned through natural selection processes. However, aphids are not common prey for ants. In fact, they are often in a mutualistic relationship with ants. It is well-known that T. bicarinatum tends aphids for their honeydew, so the aphids may not have exerted any selection pressure on the evolution of the venom chemistry of T. bicarinatum. This may be another reason why aphids are not sensitive to the venom of T. bicarinatum.
Some minor editorial changes:
• Ln 26: Change "agronomy" to "agriculture" since insect pest management is also important in livestock production and management.
• Ln 29: Do you mean "recalcitrant to degradation"?
• Ln 126: Change the caption of Figure 1 to "Neuroactivity of venom peptides by injection against three insect species."
• Ln 134: "Based on these injection assays, U11 does not appear to be relevant for agronomic development due to its neurotoxic effects on bees and its weak effect on aphids." This statement may not be appropriate since having injection toxicity does not mean the peptide is absolutely unsafe for pollinators. As your data have shown, this peptide has no oral and contact toxicity to honeybees, so it remains relevant to agriculture.
Author Response
It is a well-written paper that describes a novel structural family of neuroactive venom peptides from Tetramorium bicarinatum, one of the most successful and widespread tramp ant species. This is a significant discovery. The experiments were well-designed, the data were well-interpreted, and the conclusion is supported by the data.
1) The injected U11 caused paralysis in blowflies and honeybees but not in aphid A. pisum. The authors speculated that either the aphids are resistant to the paralytic effect of U11, or the aphids' morphology makes them less sensitive to injection than flies and bees. As the authors stated, the selectivity of ant venom on the pharmacological receptors of their prey may have been fine-tuned through natural selection processes. However, aphids are not common prey for ants. In fact, they are often in a mutualistic relationship with ants. It is well-known that T. bicarinatum tends aphids for their honeydew, so the aphids may not have exerted any selection pressure on the evolution of the venom chemistry of T. bicarinatum. This may be another reason why aphids are not sensitive to the venom of T. bicarinatum.
Thank you for this relevant comment. Based on this comment, we have added a line to the discussion line 268-270.
Some minor editorial changes:
2) Ln 26: Change "agronomy" to "agriculture" since insect pest management is also important in livestock production and management.
We have made this change.
3) Ln 29: Do you mean "recalcitrant to degradation"?
Yes, we meant persistent in the environment. We have changed "recalcitrant" to "persistent" instead to make it clearer.
4) Ln 126: Change the caption of Figure 1 to "Neuroactivity of venom peptides by injection against three insect species."
We have made this change.
5) Ln 134: "Based on these injection assays, U11 does not appear to be relevant for agronomic development due to its neurotoxic effects on bees and its weak effect on aphids." This statement may not be appropriate since having injection toxicity does not mean the peptide is absolutely unsafe for pollinators. As your data have shown, this peptide has no oral and contact toxicity to honeybees, so it remains relevant to agriculture.
That's exactly what we intended to mean. We have modified sentences lines 138, 141, and 142 and added a new sentence (lines 150-152) to clarify the statement.
Reviewer 3 Report
This is a very interesting study that examined the insecticidal effects of several ant venom peptides. One peptide toxin, U11, was characterised in more detail and showed promising oral toxicity against two dipteran species, while not being orally active against honeybees, aphids and weevils (despite being toxic to honeybees by injection). This illustrates that injection toxicity alone is no good predictor for the agronomic potential of novel bioinsecticides and that oral and/or topical application should also be examined to fully evaluate novel candidate bioinsecticides against both target and off-target species. This manuscript is of high quality and deserves publication in Toxins after addressing the minor points listed below.
Minor points:
L393-395: part of the sentence is duplicated
Was there any control of the humidity for the 72h honeybee experiments?
It's interesting to see that part of the structure of U11 is alpha-helical. Usually most insecticidal arthropod toxins conform to the ICK motif and helical structures are rare. However, Undheim et al (Undheim et al., 2015, Structure 23, 1283–1292) published the structures of helical arthropod-neuropeptide-derived (HAND) toxins from spider and centipede venoms derived from the ITP/CHH peptide family. Interestingly, the spider venom peptide Ta1a not only comprises a helical motif, but also a very potent insecticidal activity (LD50 in blowflies of 198 pMol/g). It might therefore be useful to examine and discuss potential similarities of the 3D structures of U11 with HAND peptides.
Tu support the statements made in the text, it would be beneficial to highlight those peptides that are cytotoxic in table 2.
Author Response
This is a very interesting study that examined the insecticidal effects of several ant venom peptides. One peptide toxin, U11, was characterised in more detail and showed promising oral toxicity against two dipteran species, while not being orally active against honeybees, aphids and weevils (despite being toxic to honeybees by injection). This illustrates that injection toxicity alone is no good predictor for the agronomic potential of novel bioinsecticides and that oral and/or topical application should also be examined to fully evaluate novel candidate bioinsecticides against both target and off-target species. This manuscript is of high quality and deserves publication in Toxins after addressing the minor points listed below.
Minor points:
- L393-395: part of the sentence is duplicated
Well spotted. Duplicate sentence was removed from the text.
- Was there any control of the humidity for the 72h honeybee experiments?
We were not able to control the humidity of the room for the honeybee experiments. However, the sucrose solution provided in the boxes probably helped maintain a high level of humidity inside.
- It's interesting to see that part of the structure of U11 is alpha-helical. Usually most insecticidal arthropod toxins conform to the ICK motif and helical structures are rare. However, Undheim et al (Undheim et al., 2015, Structure 23, 1283–1292) published the structures of helical arthropod-neuropeptide-derived (HAND) toxins from spider and centipede venoms derived from the ITP/CHH peptide family. Interestingly, the spider venom peptide Ta1a not only comprises a helical motif, but also a very potent insecticidal activity (LD50 in blowflies of 198 pMol/g). It might therefore be useful to examine and discuss potential similarities of the 3D structures of U11 with HAND peptides.
Yes, we are familiar with that paper. We thought of HAND toxins when we saw the U11 structure. However, while U1-agatoxin-Ta1a has a direct neurotoxic effect on the insect CNS, its exact target is not known. On the other hand, HAND toxins have 3 disulfide bridges and are not structural homologues of U11, except that all are rich in helices. Based on that comment, we added a sentence in the discussion line 249-251 but not comment further on HAND toxin structures.
- To support the statements made in the text, it would be beneficial to highlight those peptides that are cytotoxic in table 2.
Great suggestion! We have replaced the PD50 24h column in Table II, which did not add much to the discussion, with cytotoxicity information. Thanks.